# FREQUENCY PRINCIPLE: FOURIER ANALYSIS SHEDS LIGHT ON DEEP NEURAL NETWORKS

## ABSTRACT

We study the training process of Deep Neural Networks (DNNs) from the Fourier analysis perspective. We demonstrate a very universal Frequency Principle (F-Principle) — DNNs often fit target functions from low to high frequencies — on high-dimensional benchmark datasets such as MNIST/CIFAR10 and deep neural networks such as VGG16. This F-Principle of DNNs is opposite to the behavior of most conventional iterative numerical schemes (e.g., Jacobi method), which exhibit faster convergence for higher frequencies for various scientific computing problems. With theories under an idealized setting, we illustrate that this F-Principle results from the smoothness/regularity of the commonly used activation functions. The F-Principle implies an implicit bias that DNNs tend to fit training data by a low-frequency function. This understanding provides an explanation of good generalization of DNNs on most real datasets and bad generalization of DNNs on parity function or a randomized dataset.

## 1 INTRODUCTION

Understanding the training process of Deep Neural Networks (DNNs) is a fundamental problem in the area of deep learning. We find a common behavior of the gradient-based training process of DNNs, that is, a Frequency Principle (F-Principle):

*DNNs often fit target functions from low to high frequencies during the training process.*

In another word, at the early stage of training, the low-frequencies are fitted and as iteration steps of training increase, the high-frequencies are fitted. For example, when a DNN is trained to fit $y = \sin(x) + \sin(2x)$, its output would be close to $\sin(x)$ at early stage and as training goes on, its output would be close to $\sin(x) + \sin(2x)$. F-Principle was observed empirically in synthetic low-dimensional data with MSE loss during DNN training (Xu et al., 2018; Rahaman et al., 2018). However, in deep learning, empirical phenomena could vary from one network structure to another, from one dataset to another and could exhibit significant difference between synthetic data and high-dimensional real data. Therefore, the universality of the F-Principle remains an important problem for further study. Especially for high-dimensional real problems, because the computational cost of high-dimensional Fourier transform is prohibitive in practice, it is of great challenge to demonstrate the F-Principle. On the other hand, the mechanism underlying the F-Principle and its implication to the application of DNNs, e.g., design of DNN-based PDE solver, as well as their generalization ability are also important open problems to be addressed.

In this work, we design two methods, i.e., projection and filtering methods, to show that the F-Principle exists in the training process of DNNs for high-dimensional benchmarks, i.e., MNIST (LeCun, 1998), CIFAR10 (Krizhevsky et al., 2010). The settings we have considered are i) different DNN architectures, e.g., fully-connected network, convolutional neural network (CNN), and VGG16 (Simonyan & Zisserman, 2014); ii) different activation functions, e.g., tanh and rectified linear unit (ReLU); iii) different loss functions, e.g., cross entropy, mean squared error (MSE), and loss energy functional in variational problems. These results demonstrate the universality of the F-Principle.

To facilitate the designs and applications of DNN-based schemes, we characterize a stark difference between DNNs and conventional numerical schemes on various scientific computing problems, where most of the conventional methods (e.g., Jacobi method) exhibit the opposite convergence behavior

— faster convergence for higher frequencies. This difference implies that DNN can be adopted to accelerate the convergence of low frequencies for computational problems.

We also intuitively explain with theories under an idealized setting how the smoothness/regularity of commonly used activation functions contributes to the F-Principle. Note that this mechanism is rigorously demonstrated for DNNs of general settings in a subsequent work (Luo et al., 2019). Finally, we discuss that the F-Principle provides an understanding of good generalization of DNNs in many real datasets (Zhang et al., 2016) and poor generalization in learning the parity function (Shalev-Shwartz et al., 2017; Nye & Saxe, 2018), that is, the F-Principle which implies that DNNs prefer low frequencies, is consistent with the property of low frequencies dominance in many real datasets, e.g., MNIST/CIFAR10, but is different from the parity function whose spectrum concentrates on high frequencies. Compared with previous studies, our main contributions are as follows:

1. By designing both the projection and filtering methods, we consistently demonstrate the F-Principle for MNIST/CIFAR10 over various architectures such as VGG16 and various loss functions.

2. For the application of solving differential equations, we show that (i) conventional numerical schemes learn higher frequencies faster whereas DNNs learn lower frequencies faster by the F-Principle, (ii) convergence of low frequencies can be greatly accelerated with DNN-based schemes.

3. We present theories under an idealized setting to illustrate how smoothness/regularity of activation function contributes to the F-Principle.

4. We discuss in detail the implication of the F-Principle to the generalization of DNNs that DNNs are implicitly biased towards a low frequency function and provide an explanation of good and poor generalization of DNNs for low and high frequency dominant target functions, respectively.

## 2 FREQUENCY PRINCIPLE

The concept of "frequency" is central to the understanding of F-Principle. In this paper, *the "frequency" means **response frequency** NOT image (or input) frequency as explained in the following.*

Image (or input) frequency (NOT used in the paper): Frequency of 2-d function $I : \mathbb{R}^2 \to \mathbb{R}$ representing the intensity of an image over pixels at different locations. This frequency corresponds to the rate of change of intensity *across neighbouring pixels*. For example, an image of constant intensity possesses only the zero frequency, i.e., the lowest frequency, while a sharp edge contributes to high frequencies of the image.

**Response frequency** (used in the paper): Frequency of a general Input-Output mapping $f$. For example, consider a simplified classification problem of partial MNIST data using only the data with label 0 and 1, $f(x_1, x_2, \cdots, x_{784}) : \mathbb{R}^{784} \to \{0, 1\}$ mapping 784-d space of pixel values to 1-d space, where $x_j$ is the intensity of the $j$-th pixel. Denote the mapping's Fourier transform as $\hat{f}(k_1, k_2, \cdots, k_{784})$. The frequency in the coordinate $k_j$ measures the rate of change of $f(x_1, x_2, \cdots, x_{784})$ *with respect to $x_j$, i.e., the intensity of the $j$-th pixel*. If $f$ possesses significant high frequencies for large $k_j$, then a small change of $x_j$ in the image might induce a large change of the output (e.g., adversarial example). For a dataset with multiple classes, we can similarly define frequency for each output dimension. For real data, the response frequency is rigorously defined via the standard nonuniform discrete Fourier transform (NUDFT), see Appendix A.

**Frequency Principle**: *DNNs often fit target functions from low to high (response) frequencies during the training process.* An illustration of F-Principle using a function of 1-d input is in Appendix B. The F-Principle is rigorously defined through the frequency defined by the Fourier transform (Appendix A, Bracewell & Bracewell (1986)) and the converging speed defined by the relative error. By using high-dimensional real datasets, we then experimentally demonstrate F-Principle at the levels of both individual frequencies (projection method) and coarse-grained frequencies (filtering method).

## 3 F-PRINCIPLE IN MNIST/CIFAR10 THROUGH PROJECTION METHOD

Real datasets are very different from synthetic data used in previous studies. In order to utilize the F-Principle to understand and better use DNNs in real datasets, it is important to verify whether the F-Principle also holds in high-dimensional real datasets.

In the following experiments, we examine the F-Principle in a training dataset of $\{(\boldsymbol{x}_i, \boldsymbol{y}_i)\}_{i=0}^{n-1}$ where $n$ is the size of dataset. $\boldsymbol{x}_i \in \mathbb{R}^d$ is a vector representing the image and $\boldsymbol{y}_i \in \{0, 1\}^{10}$ is the output (a one-hot vector indicating the label for the dataset of image classification). $d$ is the dimension of the input ($d = 784$ for MNIST and $d = 32 \times 32 \times 3$ for CIFAR10). Since the high dimensional discrete Fourier transform (DFT) requires prohibitively high computational cost, in this section, we only consider one direction in the Fourier space through a projection method for each examination.

## 3.1 EXAMINATION METHOD: PROJECTION

For a dataset $\{(\boldsymbol{x}_i, \boldsymbol{y}_i)\}_{i=0}^{n-1}$ we consider one entry of 10-d output, denoted by $y_i \in \mathbb{R}$. The high dimensional discrete non-uniform Fourier transform of $\{(\boldsymbol{x}_i, y_i)\}_{i=0}^{n-1}$ is $\hat{y}_{\boldsymbol{k}} = \frac{1}{n} \sum_{i=0}^{n-1} y_i \exp(-\mathrm{i}2\pi\boldsymbol{k} \cdot \boldsymbol{x}_i)$. The number of all possible $\boldsymbol{k}$ grows exponentially on dimension $d$. For illustration, in each examination, we consider a direction of $\boldsymbol{k}$ in the Fourier space, i.e., $\boldsymbol{k} = k\boldsymbol{p}_1$, $\boldsymbol{p}_1$ is a chosen and fixed unit vector, hence $|\boldsymbol{k}| = k$. Then we have $\hat{y}_k = \frac{1}{n} \sum_{i=0}^{n-1} y_i \exp(-\mathrm{i}2\pi(\boldsymbol{p}_1 \cdot \boldsymbol{x}_j)k)$, which is essentially the 1-d Fourier transform of $\{(x_{\boldsymbol{p}_1,i}, y_i)\}_{i=0}^{n-1}$, where $x_{\boldsymbol{p}_1,i} = \boldsymbol{p}_1 \cdot \boldsymbol{x}_i$ is the projection of $\boldsymbol{x}_i$ on the direction $\boldsymbol{p}_1$ (Bracewell & Bracewell, 1986). For each training dataset, $\boldsymbol{p}_1$ is chosen as the first principle component of the input space. To examine the convergence behavior of different frequency components during the training, we compute the relative difference between the DNN output and the target function for selected important frequencies $k$'s at each recording step, that is, $\Delta_F(k) = |\hat{h}_k - \hat{y}_k|/|\hat{y}_k|$, where $\hat{y}_k$ and $\hat{h}_k$ are 1-d Fourier transforms of $\{y_i\}_{i=0}^{n-1}$ and the corresponding DNN output $\{h_i\}_{i=0}^{n-1}$, respectively, along $\boldsymbol{p}_1$. Note that each response frequency component, $\hat{h}_k$, of DNN output evolves as the training goes.

## 3.2 MNIST/CIFAR10

In the following, we show empirically that the F-Principle is exhibited in the selected direction during the training process of DNNs when applied to MNIST/CIFAR10 with cross-entropy loss. The network for MNIST is a fully-connected tanh DNN (784-400-200-10) and for CIFAR10 is two ReLU convolutional layers followed by a fully-connected DNN (800-400-400-400-10). All experimental details of this paper can be found in Appendix C. We consider one of the 10-d outputs in each case using non-uniform Fourier transform. As shown in Fig. 1(a) and 1(c), low frequencies dominate in both real datasets. During the training, the evolution of relative errors of certain selected frequencies (marked by black squares in Fig. 1(a) and 1(c)) is shown in Fig. 1(b) and 1(d). One can easily observe that DNNs capture low frequencies first and gradually capture higher frequencies. Clearly, this behavior is consistent with the F-Principle. For other components of the output vector and other directions of $\boldsymbol{p}$, similar phenomena are also observed.

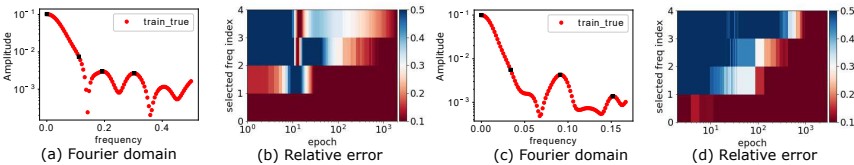

(a) Fourier domain     (b) Relative error     (c) Fourier domain     (d) Relative error

Figure 1: Projection method. (a, b) are for MNIST, (c, d) for CIFAR10. (a, c) Amplitude $|\hat{y}_k|$ vs. frequency. Selected frequencies are marked by black squares. (b, d) $\Delta_F(k)$ vs. training epochs for the selected frequencies.

## 4 F-PRINCIPLE IN MNIST/CIFAR10 THROUGH FILTERING METHOD

The projection method in the previous section enables us to visualize the F-Principle in one direction for each examination at the level of individual frequency components. However, demonstration by this method alone is insufficient because it is impossible to verify the F-Principle at all potentially informative directions for high-dimensional data. To compensate the projection method, in this

section, we consider a coarse-grained filtering method which is able to unravel whether, in the radially averaged sense, low frequencies converge faster than high frequencies.

### 4.1 EXAMINATION METHOD: FILTERING

The idea of the filtering method is as follows. We split the frequency domain into two parts, i.e., a low-frequency part with $|\boldsymbol{k}| \leq k_0$ and a high-frequency part with $|\boldsymbol{k}| > k_0$, where $|\cdot|$ is the length of a vector. The DNN is trained as usual by the original dataset $\{(\boldsymbol{x}_i, \boldsymbol{y}_i)\}_{i=0}^{n-1}$, such as MNIST or CIFAR10. The DNN output is denoted as $\boldsymbol{h}$. During the training, we can examine the convergence of relative errors of low- and high- frequency part, using the two measures below

$$e_{\text{low}} = \left( \frac{\sum_{\boldsymbol{k}} \mathbb{1}_{|\boldsymbol{k}| \leq k_0} |\hat{\boldsymbol{y}}(\boldsymbol{k}) - \hat{\boldsymbol{h}}(\boldsymbol{k})|^2}{\sum_{\boldsymbol{k}} \mathbb{1}_{|\boldsymbol{k}| \leq k_0} |\hat{\boldsymbol{y}}(\boldsymbol{k})|^2} \right)^{\frac{1}{2}}, \quad e_{\text{high}} = \left( \frac{\sum_{\boldsymbol{k}} (1 - \mathbb{1}_{|\boldsymbol{k}| \leq k_0}) |\hat{\boldsymbol{y}}(\boldsymbol{k}) - \hat{\boldsymbol{h}}(\boldsymbol{k})|^2}{\sum_{\boldsymbol{k}} (1 - \mathbb{1}_{|\boldsymbol{k}| \leq k_0}) |\hat{\boldsymbol{y}}(\boldsymbol{k})|^2} \right)^{\frac{1}{2}},$$

respectively, where $\hat{\cdot}$ indicates Fourier transform, $\mathbb{1}_{\boldsymbol{k} \leq k_0}$ is an indicator function, i.e.,

$$\mathbb{1}_{|\boldsymbol{k}| \leq k_0} = \begin{cases} 1, & |\boldsymbol{k}| \leq k_0, \\ 0, & |\boldsymbol{k}| > k_0. \end{cases}$$

If we consistently observe $e_{\text{low}} < e_{\text{high}}$ for different $k_0$'s during the training, then in a mean sense, lower frequencies are first captured by the DNN, i.e., F-Principle.

However, because it is almost impossible to compute above quantities numerically due to high computational cost of high-dimensional Fourier transform, we alternatively use the Fourier transform of a Gaussian function $\hat{G}^{\delta}(\boldsymbol{k})$, where $\delta$ is the variance of the Gaussian function $G$, to approximate $\mathbb{1}_{|\boldsymbol{k}| > k_0}$. This is reasonable due to the following two reasons. First, the Fourier transform of a Gaussian is still a Gaussian, i.e., $\hat{G}^{\delta}(\boldsymbol{k})$ decays exponentially as $|\boldsymbol{k}|$ increases, therefore, it can approximate $\mathbb{1}_{|\boldsymbol{k}| \leq k_0}$ by $\hat{G}^{\delta}(\boldsymbol{k})$ with a proper $\delta(k_0)$ (referred to as $\delta$ for simplicity). Second, the computation of $e_{\text{low}}$ and $e_{\text{high}}$ contains the multiplication of Fourier transforms in the frequency domain, which is equivalent to the Fourier transform of a convolution in the spatial domain. We can equivalently perform the examination in the spatial domain so as to avoid the almost impossible high-dimensional Fourier transform. The low frequency part can be derived by

$$\boldsymbol{y}_i^{\text{low},\delta} \triangleq (\boldsymbol{y} * G^{\delta})_i, \tag{1}$$

where $*$ indicates convolution operator, and the high frequency part can be derived by

$$\boldsymbol{y}_i^{\text{high},\delta} \triangleq \boldsymbol{y}_i - \boldsymbol{y}_i^{\text{low},\delta}. \tag{2}$$

Then, we can examine

$$e_{\text{low}} = \left( \frac{\sum_i |\boldsymbol{y}_i^{\text{low},\delta} - \boldsymbol{h}_i^{\text{low},\delta}|^2}{\sum_i |\boldsymbol{y}_i^{\text{low},\delta}|^2} \right)^{\frac{1}{2}}, \quad e_{\text{high}} = \left( \frac{\sum_i |\boldsymbol{y}_i^{\text{high},\delta} - \boldsymbol{h}_i^{\text{high},\delta}|^2}{\sum_i |\boldsymbol{y}_i^{\text{high},\delta}|^2} \right)^{\frac{1}{2}}, \tag{3}$$

where $\boldsymbol{h}^{\text{low},\delta}$ and $\boldsymbol{h}^{\text{high},\delta}$ are obtained from the DNN output $\boldsymbol{h}$, which evolves as a function of training epoch, through the same decomposition. If $e_{\text{low}} < e_{\text{high}}$ for different $\delta$'s during the training, F-Principle holds; otherwise, it is falsified. Next, we introduce the experimental procedure.

Step One: **Training**. Train the DNN by the *original dataset* $\{(\boldsymbol{x}_i, \boldsymbol{y}_i)\}_{i=0}^{n-1}$, such as MNIST or CIFAR10. $\boldsymbol{x}_i$ is an image vector, $\boldsymbol{y}_i$ is a one-hot vector.

Step Two: **Filtering**. The low frequency part can be derived by

$$\boldsymbol{y}_i^{\text{low},\delta} = \frac{1}{C_i} \sum_{j=0}^{n-1} \boldsymbol{y}_j G^{\delta}(\boldsymbol{x}_i - \boldsymbol{x}_j), \tag{4}$$

where $C_i = \sum_{j=0}^{n-1} G^{\delta}(\boldsymbol{x}_i - \boldsymbol{x}_j)$ is a normalization factor and

$$G^{\delta}(\boldsymbol{x}_i - \boldsymbol{x}_j) = \exp\left(-|\boldsymbol{x}_i - \boldsymbol{x}_j|^2 / (2\delta)\right). \tag{5}$$

The high frequency part can be derived by $\boldsymbol{y}_i^{\text{high},\delta} \triangleq \boldsymbol{y}_i - \boldsymbol{y}_i^{\text{low},\delta}$. We also compute $\boldsymbol{h}_i^{\text{low},\delta}$ and $\boldsymbol{h}_i^{\text{high},\delta}$ for each DNN output $\boldsymbol{h}_i$.

Step Three: **Examination**. To quantify the convergence of $\boldsymbol{h}^{\text{low},\delta}$ and $\boldsymbol{h}^{\text{high},\delta}$, we compute the relative error $e_{\text{low}}$ and $e_{\text{high}}$ at each training epoch through Eq. (3).

### 4.2 DNNs WITH VARIOUS SETTINGS

With the filtering method, we show the F-Principle in the DNN training process of real datasets for commonly used large networks. For MNIST, we use a fully-connected tanh-DNN (no softmax) with MSE loss; for CIFAR10, we use cross-entropy loss and two structures, one is small ReLU-CNN network, i.e., two convolutional layers, followed by a fully-connected multi-layer neural network with a softmax; the other is VGG16 (Simonyan & Zisserman, 2014) equipped with a 1024 fully-connected layer. These three structures are denoted as "DNN", "CNN" and "VGG" in Fig. 2, respectively. All are trained by SGD from *scratch*. More details are in Appendix C.

We scan a large range of $\delta$ for both datasets. As an example, results of each dataset for several $\delta$'s are shown in Fig. 2, respectively. Red color indicates small relative error. In all cases, the relative error of the low-frequency part, i.e., $e_{\text{low}}$, decreases (turns red) much faster than that of the high-frequency part, i.e., $e_{\text{high}}$. Therefore, as analyzed above, the low-frequency part converges faster than the high-frequency part. We also remark that, based on the above results on cross-entropy loss, the F-Principle is not limited to MSE loss, which possesses a natural Fourier domain interpretation by the Parseval's theorem. Note that the above results holds for both SGD and GD.

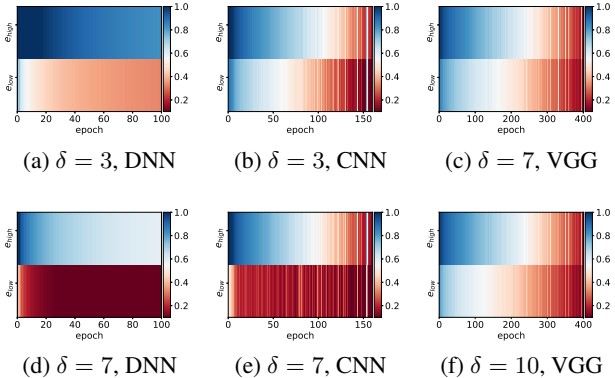

(a) $\delta = 3$, DNN      (b) $\delta = 3$, CNN      (c) $\delta = 7$, VGG

(d) $\delta = 7$, DNN      (e) $\delta = 7$, CNN      (f) $\delta = 10$, VGG

Figure 2: F-Principle in real datasets. $e_{\text{low}}$ and $e_{\text{high}}$ indicated by color against training epoch.

## 5 F-PRINCIPLE IN SOLVING DIFFERENTIAL EQUATION

Recently, DNN-based approaches have been actively explored for a variety of scientific computing problems, e.g., solving high-dimensional partial differential equations (E et al., 2017; Khoo et al., 2017; He et al., 2018; Fan et al., 2018) and molecular dynamics (MD) simulations (Han et al., 2017). However, the behaviors of DNNs applied to these problems are not well-understood. To facilitate the designs and applications of DNN-based schemes, it is important to characterize the difference between DNNs and conventional numerical schemes on various scientific computing problems. In this section, focusing on solving Poisson's equation, which has broad applications in mechanical engineering and theoretical physics (Evans, 2010), we highlight a stark difference between a DNN-based solver and the Jacobi method during the training/iteration, which can be explained by the F-Principle.

Consider a 1-d Poisson's equation:

$$-\Delta u(x) = g(x), \quad x \in \Omega \triangleq (-1, 1), \tag{6}$$
$$u(-1) = u(1) = 0. \tag{7}$$

We consider the example with $g(x) = \sin(x) + 4\sin(4x) - 8\sin(8x) + 16\sin(24x)$ which has analytic solution $u_{\text{ref}}(x) = g_0(x) + c_1 x + c_0$, where $g_0 = \sin(x) + \sin(4x)/4 - \sin(8x)/8 + \sin(24x)/36$, $c_1 = (g_0(-1) - g_0(1))/2$ and $c_0 = -(g_0(-1) + g_0(1))/2$. 1001 training samples $\{x_i\}_{i=0}^n$ are evenly spaced with grid size $\delta x$ in $[0, 1]$. Here, we use the DNN output, $h(x; \theta)$, to fit $u_{\text{ref}}(x)$ (Fig. 3(a)). A

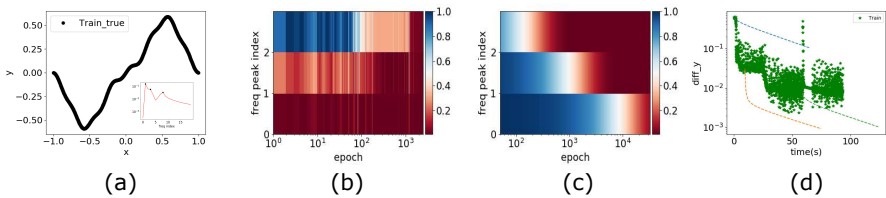

Figure 3: Poisson's equation. (a) $u_{\mathrm{ref}}(x)$. Inset: $|\hat{u}_{\mathrm{ref}}(k)|$ as a function of frequency. Frequencies peaks are marked with black dots. (b,c) $\Delta_F(k)$ computed on the inputs of training data at different epochs for the selected frequencies for DNN (b) and Jacobi (c). (d) $\|h - u_{\mathrm{ref}}\|_\infty$ at different running time. Green stars indicate $\|h - u_{\mathrm{ref}}\|_\infty$ using DNN alone. The dashed lines indicate $\|h - u_{\mathrm{ref}}\|_\infty$ for the Jacobi method with different colors indicating initialization by different timing of DNN training.

DNN-based scheme is proposed by considering the following empirical loss function (E & Yu, 2018),

$$I_{\mathrm{emp}} = \sum_{i=1}^{n-1} \left( \frac{1}{2} |\nabla_x h(x_i)|^2 - g(x_i)h(x_i) \right) \delta x + \beta \left( h(x_0)^2 + h(x_n)^2 \right). \tag{8}$$

The second term in $I_{\mathrm{emp}}(h)$ is a penalty, with constant $\beta$, arising from the Dirichlet boundary condition (7). After training, the DNN output well matches the analytical solution $u_{\mathrm{ref}}$. Focusing on the convergence of three peaks (inset of Fig. 3(a)) in the Fourier transform of $u_{\mathrm{ref}}$, as shown in Fig. 3(b), low frequencies converge faster than high frequencies as predicted by the F-Principle. For comparison, we also use the Jacobi method to solve problem (6). High frequencies converge faster in the Jacobi method (Details can be found in Appendix D), as shown in Fig. 3(c).

As a demonstration, we further propose that DNN can be combined with conventional numerical schemes to accelerate the convergence of low frequencies for computational problems. First, we solve the Poisson's equation in Eq. (6) by DNN with $M$ optimization steps (or epochs), which needs to be chosen carefully, to get a good initial guess in the sense that this solution has already learned the low frequencies (large eigenvalues) part. Then, we use the Jacobi method with the new initial data for the further iterations. We use $\|h - u_{\mathrm{ref}}\|_\infty \triangleq \max_{x \in \Omega} |h(x) - u_{\mathrm{ref}}(x)|$ to quantify the learning result. As shown by green stars in Fig. 3(d), $\|h - u_{\mathrm{ref}}\|_\infty$ fluctuates after some running time using DNN only. Dashed lines indicate the evolution of the Jacobi method with initial data set to the DNN output at the corresponding steps. If $M$ is too small (stop too early) (left dashed line), *which is equivalent to only using Jacobi*, it would take long time to converge to a small error, because low frequencies converges slowly, yet. If $M$ is too big (stop too late) (right dashed line), *which is equivalent to using DNN only*, much time would be wasted for the slow convergence of high frequencies. A proper choice of $M$ is indicated by the initial point of orange dashed line, in which low frequencies are quickly captured by the DNN, followed by fast convergence in high frequencies of the Jacobi method.

This example illustrates a cautionary tale that, although DNNs has clear advantage, using DNNs alone may not be the best option because of its limitation of slow convergence at high frequencies. Taking advantage of both DNNs and conventional methods to design faster schemes could be a promising direction in scientific computing problems.

## 6 A PRELIMINARY THEORETICAL UNDERSTANDING

A subsequent theoretical work (Luo et al., 2019) provides a rigorous mathematical study of the F-Principle at different frequencies for general DNNs (e.g., multiple hidden layers, different activation functions, high-dimensional inputs). The key insight is that the regularity of DNN converts into the decay rate of a loss function in the frequency domain. For an intuitive understanding of this key insight, we present theories under an idealized setting, which connect the smoothness/regularity of the activation function with different gradient and convergence priorities in frequency domain.

The activation function we consider is $\sigma(x) = \tanh(x)$, which is smooth in spatial domain and its derivative decays exponentially with respect to frequency in the Fourier domain. For a DNN of one

hidden layer with $m$ nodes, 1-d input $x$ and 1-d output: $h(x) = \sum_{j=1}^{m} a_j \sigma(w_j x + b_j), \quad a_j, w_j, b_j \in \mathbb{R}$. We also use the notation $\theta = \{\theta_{lj}\}$ with $\theta_{1j} = a_j$, $\theta_{2j} = w_j$, and $\theta_{3j} = b_j$, $j = 1, \cdots, m$. The loss at frequency $k$ is $L(k) = \frac{1}{2} \left| \hat{h}(k) - \hat{f}(k) \right|^2$, $\hat{\cdot}$ is the Fourier transform, $f$ is the target function. The total loss function is defined as: $L = \int_{-\infty}^{+\infty} L(k) \, \mathrm{d}k$. Note that according to Parseval's theorem, this loss function in the Fourier domain is equal to the commonly used MSE loss. We have the following theorems (The proofs are at Appendix E.). Define $W = (w_1, w_2, \cdots, w_m)^T \in \mathbb{R}^m$.

**Theorem 1.** *Considering a DNN of one hidden layer with activation function $\sigma(x) = \tanh(x)$, for any frequencies $k_1$ and $k_2$ such that $|\hat{f}(k_1)| > 0$, $|\hat{f}(k_2)| > 0$, and $|k_2| > |k_1| > 0$, there exist positive constants $c$ and $C$ such that for sufficiently small $\delta$, we have*

$$\frac{\mu\left(\left\{W : \left|\frac{\partial L(k_1)}{\partial \theta_{lj}}\right| > \left|\frac{\partial L(k_2)}{\partial \theta_{lj}}\right| \quad for \ all \quad l, j\right\} \cap B_\delta\right)}{\mu(B_\delta)} \geq 1 - C \exp(-c/\delta),$$

*where $B_\delta \subset \mathbb{R}^m$ is a ball with radius $\delta$ centered at the origin and $\mu(\cdot)$ is the Lebesgue measure.*

Theorem 1 indicates that for any two non-converged frequencies, with small weights, the lower-frequency gradient exponentially dominates over the higher-frequency ones. Due to Parseval's theorem, the MSE loss in the spatial domain is equivalent to the L2 loss in the Fourier domain. To intuitively understand the higher decay rate of a lower-frequency loss function, we consider the training in the Fourier domain with loss function of only two non-zero frequencies.

**Theorem 2.** *Considering a DNN of one hidden layer with activation function $\sigma(x) = \tanh(x)$. Suppose the target function has only two non-zero frequencies $k_1$ and $k_2$, that is, $|\hat{f}(k_1)| > 0$, $|\hat{f}(k_2)| > 0$, $|k_2| > |k_1| > 0$, and $|\hat{f}(k)| = 0$ for $k \neq k_1, k_2$. Consider the loss function of $L = L(k_1) + L(k_2)$ with gradient descent training. Denote*

$$\mathcal{S} = \left\{ \frac{\partial L(k_1)}{\partial t} \leq 0, \frac{\partial L(k_1)}{\partial t} \leq \frac{\partial L(k_2)}{\partial t} \right\},$$

*that is, $L(k_1)$ decreases faster than $L(k_2)$. There exist positive constants $c$ and $C$ such that for sufficiently small $\delta$, we have*

$$\frac{\mu\left(\{W : \mathcal{S} \quad holds\} \cap B_\delta\right)}{\mu(B_\delta)} \geq 1 - C \exp(-c/\delta),$$

*where $B_\delta \subset \mathbb{R}^m$ is a ball with radius $\delta$ centered at the origin and $\mu(\cdot)$ is the Lebesgue measure.*

# 7 DISCUSSIONS

DNNs often generalize well for real problems (Zhang et al., 2016) but poorly for problems like fitting a parity function (Shalev-Shwartz et al., 2017; Nye & Saxe, 2018) despite excellent training accuracy for all problems. Understanding the differences between above two types of problems, i.e., good and bad generalization performance of DNN, is critical. In the following, we show a qualitative difference between these two types of problems through *Fourier analysis* and use the *F-Principle* to provide an explanation different generalization performances of DNNs.

For MNIST/CIFAR10, we examine $\hat{y}_{\text{total},\boldsymbol{k}} = \frac{1}{n_{\text{total}}} \sum_{i=0}^{n_{\text{total}}-1} y_i \exp\left(-\mathrm{i}2\pi\boldsymbol{k} \cdot \boldsymbol{x}_i\right)$, where $\{(\boldsymbol{x}_i, y_i)\}_{i=0}^{n_{\text{total}}-1}$ consists of both the training and test datasets with certain selected output component, at different directions of $\boldsymbol{k}$ in the Fourier space. We find that $\hat{y}_{\text{total},\boldsymbol{k}}$ concentrates on the low frequencies along those examined directions. For illustration, $\hat{y}_{\text{total},\boldsymbol{k}}$'s along the first principle component are shown by green lines in Fig. 4(a, b) for MNIST/CIFAR10, respectively. When only the training dataset is used, $\hat{y}_{\text{train},\boldsymbol{k}}$ well overlaps with $\hat{y}_{\text{total},\boldsymbol{k}}$ at the dominant low frequencies.

For the parity function $f(\boldsymbol{x}) = \prod_{j=1}^{d} x_j$ defined on $\Omega = \{-1, 1\}^d$, its Fourier transform is $\hat{f}(\boldsymbol{k}) = \frac{1}{2^d} \sum_{\boldsymbol{x} \in \Omega} \prod_{j=1}^{d} x_j \mathrm{e}^{-\mathrm{i}2\pi\boldsymbol{k}\cdot\boldsymbol{x}} = (-\mathrm{i})^d \prod_{j=1}^{d} \sin 2\pi k_j$. Clearly, for $\boldsymbol{k} \in [-\frac{1}{4}, \frac{1}{4}]^d$, the power of the parity function concentrates at $\boldsymbol{k} \in \{-\frac{1}{4}, \frac{1}{4}\}^d$ and vanishes as $\boldsymbol{k} \to \boldsymbol{0}$, as illustrated in Fig. 4(c) for the direction of $\boldsymbol{1}_d$. Given a randomly sampled training dataset $S \subset \Omega$ with $s$ points, the nonuniform

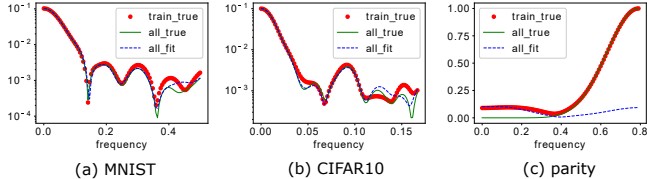

(a) MNIST        (b) CIFAR10        (c) parity

Figure 4: Fourier analysis for different generalization ability. The plot is the amplitude of the Fourier coefficient against frequency $k$. The red dots are for the training dataset, the green line is for the whole dataset, and the blue dashed line is for an output of well-trained DNN on the input of the whole dataset. For (c), $d = 10$. The training data is 200 randomly selected points.

Fourier transform on $S$ is computed as $\hat{f}_S(\boldsymbol{k}) = \frac{1}{s} \sum_{x \in S} \prod_{j=1}^{d} x_j \mathrm{e}^{-\mathrm{i}2\pi \boldsymbol{k} \cdot \boldsymbol{x}}$. As shown in Fig. 4(c), $\hat{f}(\boldsymbol{k})$ and $\hat{f}_S(\boldsymbol{k})$ significantly differ at low frequencies.

By experiments, the generalization ability of DNNs can be well reflected by the Fourier analysis. For the MNIST/CIFAR10, we observed the Fourier transform of the output of a well-trained DNN on $\{\boldsymbol{x}_i\}_{i=0}^{n_{\mathrm{total}}-1}$ faithfully recovers the dominant low frequencies, as illustrated in Fig. 4(a) and 4(b), respectively, indicating a good generalization performance as observed in experiments. However, for the parity function, we observed that the Fourier transform of the output of a well-trained DNN on $\{\boldsymbol{x}_i\}_{i \in S}$ significantly deviates from $\hat{f}(\boldsymbol{k})$ at almost all frequencies, as illustrated in Fig. 4(c), indicating a bad generalization performance as observed in experiments.

The F-Principle implicates that among all the functions that can fit the training data, a DNN is implicitly biased during the training towards a function with more power at low frequencies. If the target function has significant high-frequency components, insufficient training samples will lead to artificial low frequencies in training dataset (see red line in Fig. 4(c)), which is the well-known *aliasing* effect. Based on the F-Principle, as demonstrated in Fig. 4(c), these artificial low frequency components will be first captured to explain the training samples, whereas the high frequency components will be compromised by DNN. For MNIST/CIFAR10, since the power of high frequencies is much smaller than that of low frequencies, artificial low frequencies caused by aliasing can be neglected. To conclude, the distribution of power in Fourier domain of above two types of problems exhibits significant differences, which result in different generalization performances of DNNs according to the F-Principle.

# 8 RELATED WORK

There are different approaches attempting to explain why DNNs often generalize well. For example, generalization error is related to various complexity measures (Bartlett et al., 1999; Neyshabur et al., 2017; E et al., 2018), local properties (sharpness/flatness) of loss functions at minima (Keskar et al., 2016; Wu et al., 2017), stability of optimization algorithms (Hardt et al., 2015), and implicit bias of the training process (Soudry et al., 2018; Arpit et al., 2017; Xu et al., 2018). On the other hand, several works focus on the failure of DNNs (Shalev-Shwartz et al., 2017; Nye & Saxe, 2018), e.g., fitting the parity function, in which a well-trained DNN possesses no generalization ability. We propose that the Fourier analysis can provide insights into both success and failure of DNNs.

F-Principle was first discovered in (Xu et al., 2018; Rahaman et al., 2018) simultaneously through simple synthetic data and not very deep networks. In the revised version, Rahaman et al. (2018) examines the F-Principle in the MNIST dataset. However, they add noise to MNIST, which contaminates the labels and damages the structure of real data. They only examine not very deep (6-layer) fully connected ReLU network with MSE loss, while cross-entropy loss is widely used. This paper verified that F-Principle holds in the training process of MNIST and CIFAR10, both CNN and fully connected networks, very deep networks (VGG16) and various loss functions, e.g., MSE Loss, cross-entropy loss and variational loss function. In the aspect of theoretical study, based on the key mechanism found by the theoretical study in this paper, Luo et al. (2019) shows a rigorous proof of the F-Principle for general DNNs. The theoretical study of the gradient of $\tanh(x)$ in the Fourier domain is adopted

by Rahaman et al. (2018), in which they generalize the analysis to ReLU and show similar results. Thm 1 is also used to analyze a nonlinear collaborative scheme for deep network training (Zhen et al., 2018). In the aspect of application, based on the study of the F-Principle in this paper, Cai et al. (2019) and Cai & Xu (2019) design DNN-based algorithms to solve high-dimensional and high-frequency problems.

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

# A  RESPONSE FREQUENCY OF TRAINING DATA $\{y_i\}_{i=0}^{n-1}$ ON INPUTS $\{x_i\}_{i=0}^{n-1}$

In all our experiments, we consistently consider the response frequency defined for the mapping function $g$ between inputs and outputs, say $\mathbb{R}^d \to \mathbb{R}$ and any $\boldsymbol{k} \in \mathbb{R}^d$ via the standard nonuniform discrete Fourier transform (NUDFT)

$$\hat{g}_{\boldsymbol{k}} = \frac{1}{n} \sum_{i=0}^{n-1} g(\boldsymbol{x}_i) \mathrm{e}^{-\mathrm{i}2\pi\boldsymbol{k}\cdot\boldsymbol{x}_i},$$

which is a natural estimator of frequency composition of $g$. (More details can be found in `https://en.wikipedia.org/wiki/Non-uniform_discrete_Fourier_transform`.) As $n \to \infty$, $\hat{g}_{\boldsymbol{k}} \to \int g(\boldsymbol{x})\mathrm{e}^{-\mathrm{i}2\pi\boldsymbol{k}\cdot\boldsymbol{x}}\nu(\boldsymbol{x})\,\mathrm{d}\boldsymbol{x}$, where $\nu(\boldsymbol{x})$ is the data distribution.

We restrict all the evaluation of Fourier transform in our experiments to NUDFT of $\{y_i\}_{i=0}^{n-1}$ at $\{x_i\}_{i=0}^{n-1}$ for the following practical reasons.

(i) The information of target function is only available at $\{x_i\}_{i=0}^{n-1}$ for training.

(ii) It allows us to perform the convergence analysis. As $t \to \infty$, in general, $h(\boldsymbol{x}_i, t) \to \boldsymbol{y}_i$ for any $i$ ($h(\boldsymbol{x}_i, t)$ is the DNN output), leading to $\hat{h}_{\boldsymbol{k}} \to \hat{y}_{\boldsymbol{k}}$ for any $\boldsymbol{k}$. Therefore, we can analyze the convergence at different $\boldsymbol{k}$ by evaluating $\Delta_F(\boldsymbol{k}) = |\hat{h}_{\boldsymbol{k}} - \hat{y}_{\boldsymbol{k}}|/|\hat{y}_{\boldsymbol{k}}|$ during the training. If we use a different set of data points for frequency evaluation of DNN output, then $\Delta_F(\boldsymbol{k})$ may not converge to $0$ at the end of training.

(iii) $\hat{y}_{\boldsymbol{k}}$ faithfully reflect the frequency structure of training data $\{x_i, y_i\}_{i=0}^{n-1}$. Intuitively, high frequencies of $\hat{y}_{\boldsymbol{k}}$ correspond to sharp changes of output for some nearby points in the training data. Then, by applying a Gaussian filter and evaluating still at $\{x_i\}_{i=0}^{n-1}$, we obtain the low frequency part of training data with these sharp changes (high frequencies) well suppressed.

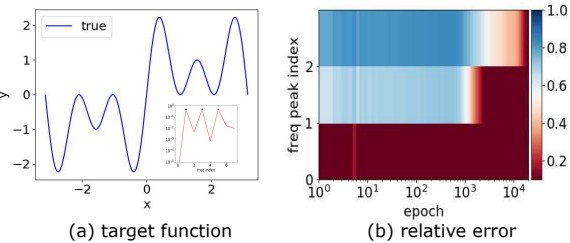

(a) target function        (b) relative error

Figure 5: 1d input. (a) $f(x)$. Inset : $|\hat{f}(k)|$. (b) $\Delta_F(k)$ of three important frequencies (indicated by black dots in the inset of (a)) against different training epochs.

## B  ILLUSTRATION OF F-PRINCIPLE FOR 1-D SYNTHETIC DATA

To illustrate the phenomenon of F-Principle, we use 1-d synthetic data to show the evolution of relative training error at different frequencies during the training of DNN. we train a DNN to fit a 1-d target function $f(x) = \sin(x) + \sin(3x) + \sin(5x)$ of three frequency components. On $n = 201$ evenly spaced training samples, i.e., $\{x_i\}_{i=0}^{n-1}$ in $[-3.14, 3.14]$, the discrete Fourier transform (DFT) of $f(x)$ or the DNN output (denoted by $h(x)$) is computed by $\hat{f}_k = \frac{1}{n}\sum_{i=0}^{n-1} f(x_i)e^{-i2\pi ik/n}$ and $\hat{h}_k = \frac{1}{n}\sum_{i=0}^{n-1} h(x_i)e^{-i2\pi jk/n}$, where $k$ is the frequency. As shown in Fig. 5(a), the target function has three important frequencies as we design (black dots at the inset in Fig. 5(a)). To examine the convergence behavior of different frequency components during the training with MSE, we compute the relative difference between the DNN output and the target function for the three important frequencies $k$'s at each recording step, that is, $\Delta_F(k) = |\hat{h}_k - \hat{f}_k|/|\hat{f}_k|$, where $|\cdot|$ denotes the norm of a complex number. As shown in Fig. 5(b), the DNN converges the first frequency peak very fast, while converging the second frequency peak much slower, followed by the third frequency peak.

Next, we investigate the F-Principle on real datasets with more general loss functions other than MSE which was the only loss studied in the previous works (Xu et al., 2018; Rahaman et al., 2018). All experimental details can be found in Appendix. C.

## C  EXPERIMENTAL SETTINGS

In Fig. 5, the parameters of the DNN is initialized by a Gaussian distribution with mean 0 and standard deviation 0.1. We use a tanh-DNN with widths 1-8000-1 with full batch training. The learning rate is 0.0002. The DNN is trained by Adam optimizer (Kingma & Ba, 2014) with the MSE loss function.

In Fig. 1, for MNIST dataset, the training process of a tanh-DNN with widths 784-400-200-10 is shown in Fig. 1(a) and 1(b). For CIFAR10 dataset, results are shown in Fig. 1(c) and 1(d) of a ReLU-CNN, which consists of one convolution layer of $3 \times 3 \times 64$, a max pooling of $2 \times 2$, one convolution layer of $3 \times 3 \times 128$, a max pooling of $2 \times 2$, followed by a fully-connected DNN with widths 800-400-400-400-10. For both cases, the output layer of the network is equipped with a softmax. The network output is a 10-d vector. The DNNs are trained with cross entropy loss by Adam optimizer (Kingma & Ba, 2014). (a, b) are for MNIST with a tanh-DNN. The learning rate is 0.001 with batch size 10000. After training, the training accuracy is 0.951 and test accuracy is 0.963. The amplitude of the Fourier coefficient with respect to the fourth output component at each frequency is shown in (a), in which the red dots are computed using the training data. Selected frequencies are marked by black squares. (b) $\Delta_F(k)$ at different training epochs for the selected frequencies. (c, d)

are for CIFAR10 dataset. We use a ReLU network of a CNN followed by a fully-connected DNN. The learning rate is $0.003$ with batch size $512$. (c) and (d) are the results with respect to the ninth output component. After training, the training accuracy is $0.98$ and test accuracy is $0.72$.

In Fig. 2, for MNIST, we use a fully-connected tanh-DNN with widths 784-400-200-10 and MSE loss; for CIFAR10, we use cross-entropy loss and a ReLU-CNN, which consists of one convolution layer of $3 \times 3 \times 32$, a max pooling of $2 \times 2$, one convolution layer of $3 \times 3 \times 64$, a max pooling of $2 \times 2$, followed by a fully-connected DNN with widths 400-10 and the output layer of the network is equipped with a softmax. The learning rate for MNIST and CIFAR10 is $0.015$ and $0.003$, respectively. The networks are trained by Adam optimizer (Kingma & Ba, 2014) with batch size 10000. For VGG16, the learning rate is $10^{-5}$. The network is trained by Adam optimizer (Kingma & Ba, 2014) with batch size 500.

In Fig. 3, the samples are evenly spaced in $[0, 1]$ with sample size 1001. We use a DNN with widths 1-4000-500-400-1 and full batch training by Adam optimizer (Kingma & Ba, 2014). The learning rate is $0.0005$. $\beta$ is $10$. The parameters of the DNN are initialized following a Gaussian distribution with mean 0 and standard deviation 0.02.

In Fig. 4, the settings of (a) and (b) are the same as the ones in Fig. 1. For (c), we use a tanh-DNN with widths 10-500-100-1, learning rate $0.0005$ under full batch-size training by Adam optimizer (Kingma & Ba, 2014). The parameters of the DNN are initialized by a Gaussian distribution with mean 0 and standard deviation 0.05.

## D    CENTRAL DIFFERENCE SCHEME AND JACOBI METHOD

Consider a one-dimensional (1-d) Poisson's equation:

$$- \Delta u(x) = g(x), \quad x \in \Omega = (-1, 1) \tag{9}$$

$$u(x) = 0, \quad x = -1, 1.$$

$[-1, 1]$ is uniformly discretized into $n + 1$ points with grid size $h = 2/n$. The Poisson's equation in Eq. (9) can be solved by the central difference scheme,

$$- \Delta u_i = - \frac{u_{i+1} - 2u_i + u_{i-1}}{(\delta x)^2} = g(x_i), \quad i = 1, 2, \cdots, n, \tag{10}$$

resulting a linear system

$$\boldsymbol{A}\boldsymbol{u} = \boldsymbol{g}, \tag{11}$$

where

$$\boldsymbol{A} = \begin{pmatrix} 2 & -1 & 0 & 0 & \cdots & 0 \\ -1 & 2 & -1 & 0 & \cdots & 0 \\ 0 & -1 & 2 & -1 & \cdots & 0 \\ \vdots & \vdots & \cdots & & & \vdots \\ 0 & 0 & \cdots & 0 & -1 & 2 \end{pmatrix}_{(n-1) \times (n-1)}, \tag{12}$$

$$\boldsymbol{u} = \begin{pmatrix} u_1 \\ u_2 \\ \vdots \\ u_{n-2} \\ u_{n-1} \end{pmatrix}, \quad \boldsymbol{g} = (\delta x)^2 \begin{pmatrix} g_1 \\ g_2 \\ \vdots \\ g_{n-2} \\ g_{n-1} \end{pmatrix}, \quad x_i = 2\frac{i}{n}. \tag{13}$$

A class of methods to solve this linear system is iterative schemes, for example, the Jacobi method. Let $\boldsymbol{A} = \boldsymbol{D} - \boldsymbol{L} - \boldsymbol{U}$, where $\boldsymbol{D}$ is the diagonal of $\boldsymbol{A}$, and $\boldsymbol{L}$ and $\boldsymbol{U}$ are the strictly lower and upper triangular parts of $-\boldsymbol{A}$, respectively. Then, we obtain

$$\boldsymbol{u} = \boldsymbol{D}^{-1}(\boldsymbol{L} + \boldsymbol{U})\boldsymbol{u} + \boldsymbol{D}^{-1}\boldsymbol{g}. \tag{14}$$

At step $t \in \mathbb{N}$, the Jacobi iteration reads as

$$\boldsymbol{u}^{t+1} = \boldsymbol{D}^{-1}(\boldsymbol{L} + \boldsymbol{U})\boldsymbol{u}^t + \boldsymbol{D}^{-1}\boldsymbol{g}. \tag{15}$$

We perform the standard error analysis of the above iteration process. Denote $\boldsymbol{u}^*$ as the true value obtained by directly performing inverse of $\boldsymbol{A}$ in Eq. (11). The error at step $t+1$ is $\boldsymbol{e}^{t+1} = \boldsymbol{u}^{t+1} - \boldsymbol{u}^*$. Then, $\boldsymbol{e}^{t+1} = \boldsymbol{R}_J \boldsymbol{e}^t$, where $\boldsymbol{R}_J = \boldsymbol{D}^{-1}(\boldsymbol{L} + \boldsymbol{U})$. The converging speed of $\boldsymbol{e}^t$ is determined by the eigenvalues of $\boldsymbol{R}_J$, that is,

$$\lambda_k = \lambda_k(\boldsymbol{R}_J) = \cos\frac{k\pi}{n}, \quad k = 1, 2, \cdots, n-1, \tag{16}$$

and the corresponding eigenvector $\boldsymbol{v}_k$'s entry is

$$v_{k,i} = \sin\frac{ik\pi}{n}, i = 1, 2, \cdots, n-1. \tag{17}$$

So we can write

$$\boldsymbol{e}^t = \sum_{k=1}^{n-1} \alpha_k^t \boldsymbol{v}_k, \tag{18}$$

where $\alpha_k^t$ can be understood as the magnitude of $\boldsymbol{e}^t$ in the direction of $\boldsymbol{v}_k$. Then,

$$\boldsymbol{e}^{t+1} = \sum_{k=1}^{n-1} \alpha_k^t \boldsymbol{R}_J \boldsymbol{v}_k = \sum_{k=1}^{n-1} \alpha_k^t \lambda_k \boldsymbol{v}_k. \tag{19}$$

$$\alpha_k^{t+1} = \lambda_k \alpha_k^t.$$

Therefore, the converging rate of $\boldsymbol{e}^t$ in the direction of $\boldsymbol{v}_k$ is controlled by $\lambda_k$. Since

$$\cos\frac{k\pi}{n} = -\cos\frac{(n-k)\pi}{n}, \tag{20}$$

the frequencies $k$ and $(n-k)$ are closely related and converge with the same rate. Consider the frequency $k < n/2$, $\lambda_k$ is larger for lower frequency. Therefore, lower frequency converges slower in the Jacobi method.

## E PROOF OF THEOREMS

The activation function we consider is $\sigma(x) = \tanh(x)$.

$$\sigma(x) = \tanh(x) = \frac{e^x - e^{-x}}{e^x + e^{-x}}, \quad x \in \mathbb{R}.$$

For a DNN of one hidden layer with $m$ nodes, 1-d input $x$ and 1-d output:

$$h(x) = \sum_{j=1}^{m} a_j \sigma(w_j x + b_j), \quad a_j, w_j, b_j \in \mathbb{R}, \tag{21}$$

where $w_j$, $a_j$, and $b_j$ are called *parameters*, in particular, $w_j$ and $a_j$ are called *weights*, and $b_j$ is also known as a *bias*. In the sequel, we will also use the notation $\theta = \{\theta_{lj}\}$ with $\theta_{1j} = a_j$, $\theta_{2j} = w_j$, and $\theta_{lj} = b_j$, $j = 1, \cdots, m$. Note that $\hat{\sigma}(k) = -\frac{i\pi}{\sinh(\pi k/2)}$ where the Fourier transformation and its inverse transformation are defined as follows:

$$\hat{f}(k) = \int_{-\infty}^{+\infty} f(x)e^{-ikx}\, dx, \quad f(x) = \frac{1}{2\pi}\int_{-\infty}^{+\infty} \hat{f}(k)e^{ikx}\, dk.$$

The Fourier transform of $\sigma(w_j x + b_j)$ with $w_j, b_j \in \mathbb{R}$, $j = 1, \cdots, m$ reads as

$$\widehat{\sigma(w_j \cdot + b_j)}(k) = \frac{2\pi i}{|w_j|}\exp\left(\frac{ib_j k}{w_j}\right)\frac{1}{\exp(-\frac{\pi k}{2w_j}) - \exp(\frac{\pi k}{2w_j})}. \tag{22}$$

Thus

$$\hat{h}(k) = \sum_{j=1}^{m} \frac{2\pi a_j i}{|w_j|}\exp\left(\frac{ib_j k}{w_j}\right)\frac{1}{\exp(-\frac{\pi k}{2w_j}) - \exp(\frac{\pi k}{2w_j})}. \tag{23}$$

We define the amplitude deviation between DNN output and the *target function* $f(x)$ at frequency $k$ as

$$D(k) \triangleq \hat{h}(k) - \hat{f}(k).$$

Write $D(k)$ as $D(k) = A(k)e^{i\phi(k)}$, where $A(k) \in [0, +\infty)$ and $\phi(k) \in \mathbb{R}$ are the amplitude and phase of $D(k)$, respectively. The loss at frequency $k$ is $L(k) = \frac{1}{2}|D(k)|^2$, where $|\cdot|$ denotes the norm of a complex number. The total loss function is defined as: $L = \int_{-\infty}^{+\infty} L(k)\,dk$. Note that according to Parseval's theorem, this loss function in the Fourier domain is equal to the commonly used loss of mean squared error, that is, $L = \int_{-\infty}^{+\infty} \frac{1}{2}(h(x) - f(x))^2\,dx$. For readers' reference, we list the partial derivatives of $L(k)$ with respect to parameters

$$\frac{\partial L(k)}{\partial a_j} = \frac{2\pi}{w_j}\sin\left(\frac{b_j k}{w_j} - \phi(k)\right)E_0, \tag{24}$$

$$\frac{\partial L(k)}{\partial w_j} = \left[\sin\left(\frac{b_j k}{w_j} - \phi(k)\right)\left(\frac{\pi^2 a_j k}{w_j^3}E_1 - \frac{2\pi a_j}{w_j^2}\right)\right. $$
$$\left. - \frac{2\pi a_j b_j k}{w_j^3}\cos\left(\frac{b_j k}{w_j} - \phi(k)\right)\right]E_0, \tag{25}$$

$$\frac{\partial L(k)}{\partial b_j} = \frac{2\pi a_j b_j k}{w_j^2}\cos\left(\frac{b_j k}{w_j} - \phi(k)\right)E_0, \tag{26}$$

where

$$E_0 = \frac{\mathrm{sgn}(w_j)A(k)}{\exp(\frac{\pi k}{2w_j}) - \exp(-\frac{\pi k}{2w_j})},$$

$$E_1 = \frac{\exp(\frac{\pi k}{2w_j}) + \exp(-\frac{\pi k}{2w_j})}{\exp(\frac{\pi k}{2w_j}) - \exp(-\frac{\pi k}{2w_j})}.$$

The descent increment at any direction, say, with respect to parameter $\theta_{lj}$, is

$$\frac{\partial L}{\partial \theta_{lj}} = \int_{-\infty}^{+\infty} \frac{\partial L(k)}{\partial \theta_{lj}}\,dk. \tag{27}$$

The absolute contribution from frequency $k$ to this total amount at $\theta_{lj}$ is

$$\left|\frac{\partial L(k)}{\partial \theta_{lj}}\right| \approx A(k)\exp\left(-|\pi k/2w_j|\right)F_{lj}(\theta_j, k), \tag{28}$$

where $\theta_j \triangleq \{w_j, b_j, a_j\}$, $\theta_{lj} \in \theta_j$, $F_{lj}(\theta_j, k)$ is a function with respect to $\theta_j$ and $k$, which can be found in one of Eqs. (24, 25, 26).

When the component at frequency $k$ where $\hat{h}(k)$ is not close enough to $\hat{f}(k)$, $\exp\left(-|\pi k/2w_j|\right)$ would dominate $G_{lj}(\theta_j, k)$ for a small $w_j$. Through the above framework of analysis, we have the following theorem. Define

$$W = (w_1, w_2, \cdots, w_m)^T \in \mathbb{R}^m. \tag{29}$$

**Theorem.** *Consider a one hidden layer DNN with activation function $\sigma(x) = \tanh x$. For any frequencies $k_1$ and $k_2$ such that $|\hat{f}(k_1)| > 0$, $|\hat{f}(k_2)| > 0$, and $|k_2| > |k_1| > 0$, there exist positive constants $c$ and $C$ such that for sufficiently small $\delta$, we have*

$$\frac{\mu\left(\left\{W : \left|\frac{\partial L(k_1)}{\partial \theta_{lj}}\right| > \left|\frac{\partial L(k_2)}{\partial \theta_{lj}}\right| \quad for \ all \quad l, j\right\} \cap B_\delta\right)}{\mu(B_\delta)}$$
$$\geq 1 - C\exp(-c/\delta), \tag{30}$$

*where $B_\delta \subset \mathbb{R}^m$ is a ball with radius $\delta$ centered at the origin and $\mu(\cdot)$ is the Lebesgue measure.*

We remark that $c$ and $C$ depend on $k_1$, $k_2$, $|\hat{f}(k_1)|$, $|\hat{f}(k_2)|$, $\sup|a_i|$, $\sup|b_i|$, and $m$.

*Proof.* To prove the statement, it is sufficient to show that $\mu(S_{lj,\delta})/\mu(B_\delta) \leq C\exp(-c/\delta)$ for each $l, j$, where

$$S_{lj,\delta} := \left\{ W \in B_\delta : \left|\frac{\partial L(k_1)}{\partial \theta_{lj}}\right| \leq \left|\frac{\partial L(k_2)}{\partial \theta_{lj}}\right| \right\}. \tag{31}$$

We prove this for $S_{1j,\delta}$, that is, $\theta_{lj} = a_j$. The proofs for $\theta_{lj} = w_j$ and $b_j$ are similar. Without loss of generality, we assume that $k_1, k_2 > 0$, $b_j > 0$, and $w_j \neq 0$, $j = 1, \cdots, m$. According to Eq. (24), the inequality $|\frac{\partial L(k_1)}{\partial a_j}| \leq |\frac{\partial L(k_2)}{\partial a_j}|$ is equivalent to

$$\frac{A(k_2)}{A(k_1)} \left|\frac{\exp(\frac{\pi k_1}{2w_j}) - \exp(-\frac{\pi k_1}{2w_j})}{\exp(\frac{\pi k_2}{2w_j}) - \exp(-\frac{\pi k_2}{2w_j})}\right| \cdot \left|\sin\left(\frac{b_j k_2}{w_j} - \phi(k_2)\right)\right| \geq \left|\sin\left(\frac{b_j k_1}{w_j} - \phi(k_1)\right)\right| \tag{32}$$

Note that $|\hat{h}(k)| \leq C\sum_{j=1}^m \frac{|a_j|}{|w_j|}\exp(-\frac{\pi k}{2|w_j|})$ for $k > 0$. Thus

$$\lim_{W\to 0} \hat{h}(k) = 0 \quad \text{and} \quad \lim_{W\to 0} D(k) = -\hat{f}(k). \tag{33}$$

Therefore,

$$\lim_{W\to 0} A(k) = |\hat{f}(k)| \quad \text{and} \quad \lim_{W\to 0} \phi(k) = \pi + \arg(\hat{f}(k)). \tag{34}$$

For $W \in B_\delta$ with sufficiently small $\delta$, $A(k_1) > \frac{1}{2}|\hat{f}(k_1)| > 0$ and $A(k_2) < 2|\hat{f}(k_2)|$. Also note that $|\sin(\frac{b_j k_2}{w_j} - \phi(k_2))| \leq 1$ and that for sufficiently small $\delta$,

$$\left|\frac{\exp(\frac{\pi k_1}{2w_j}) - \exp(-\frac{\pi k_1}{2w_j})}{\exp(\frac{\pi k_2}{2w_j}) - \exp(-\frac{\pi k_2}{2w_j})}\right| \leq 2\exp\left(\frac{-\pi(k_2 - k_1)}{2|w_j|}\right). \tag{35}$$

Thus, inequality (32) implies that

$$\left|\sin\left(\frac{b_j k_1}{w_j} - \phi(k_1)\right)\right| \leq \frac{8|\hat{f}(k_2)|}{|\hat{f}(k_1)|}\exp\left(-\frac{\pi(k_2 - k_1)}{2|w_j|}\right). \tag{36}$$

Noticing that $\frac{2}{\pi}|x| \leq |\sin x|$ ($|x| \leq \frac{\pi}{2}$) and Eq. (34), we have for $W \in S_{lj,\delta}$, for some $q \in \mathbb{Z}$,

$$\left|\frac{b_i k_1}{w_i} - \arg(\hat{f}(k_1)) - q\pi\right| \leq \frac{8\pi|\hat{f}(k_2)|}{|\hat{f}(k_1)|}\exp\left(-\frac{\pi(k_2 - k_1)}{2\delta}\right) \tag{37}$$

that is,

$$-c_1\exp(-c_2/\delta) + q\pi + \arg(\hat{f}(k_1)) \leq \frac{b_i k_1}{w_i} \leq c_1\exp(-c_2/\delta) + q\pi + \arg(\hat{f}(k_1)), \tag{38}$$

where $c_1 = \frac{8\pi|\hat{f}(k_2)|}{|\hat{f}(k_1)|}$ and $c_2 = \pi(k_2 - k_1)$. Define $I := I^+ \cup I^-$ where

$$I^+ := \{w_j > 0 : W \in S_{1j,\delta}\}, \quad I^- := \{w_j < 0 : W \in S_{1j,\delta}\}. \tag{39}$$

For $w_j > 0$, we have for some $q \in \mathbb{Z}$,

$$0 < \frac{b_j k_1}{c_1\exp(-c_2/\delta) + q\pi + \arg(\hat{f}(k_1))} \leq w_j \leq \frac{b_j k_1}{-c_1\exp(-c_2/\delta) + q\pi + \arg(\hat{f}(k_1))}. \tag{40}$$

Since $W \in B_\delta$ and $c_1\exp(-c_2/\delta) + \arg(\hat{f}(k_1)) \leq 2\pi$, we have $\frac{b_j k_1}{2\pi + q\pi} \leq w_j \leq \delta$. Then Eq. (40) only holds for some large $q$, more precisely, $q \geq q_0 := \frac{b_j k}{\pi \delta} - 2$. Thus we obtain the estimate for the (one-dimensional) Lebesgue measure of $I^+$

$$\mu(I^+) \leq \sum_{q=q_0}^\infty \left|\frac{b_j k_1}{-c_1\exp(-c_2/\delta) + q\pi + \arg(\hat{f}(k_1))} - \frac{b_j k_1}{c_1\exp(-c_2/\delta) + q\pi + \arg(\hat{f}(k_1))}\right|$$

$$\leq 2|b_j|k_1 c_1\exp(-c_2/\delta) \cdot \sum_{q=q_0}^\infty \frac{1}{(q\pi + \arg(\hat{f}(k_1)))^2 - (c_1\exp(-c_2/\delta))^2}$$

$$\leq C\exp(-c/\delta). \tag{41}$$

The similar estimate holds for $\mu(I^-)$, and hence $\mu(I) \leq C \exp(-c/\delta)$. For $W \in B_\delta$, the $(m-1)$ dimensional vector $(w_1, \cdots, w_{j-1}, w_{j+1}, \cdots, w_m)^T$ is in a ball with radius $\delta$ in $\mathbb{R}^{m-1}$. Therefore, we final arrive at the desired estimate

$$\frac{\mu(S_{1j,\delta})}{\mu(B_\delta)} \leq \frac{\mu(I)\omega_{m-1}\delta^{m-1}}{\omega_m\delta^m} \leq C \exp(-c/\delta), \tag{42}$$

where $\omega_m$ is the volume of a unit ball in $\mathbb{R}^m$. $\qquad\square$

**Theorem.** *Considering a DNN of one hidden layer with activation function $\sigma(x) = \tanh(x)$. Suppose the target function has only two non-zero frequencies $k_1$ and $k_2$, that is, $|\hat{f}(k_1)| > 0$, $|\hat{f}(k_2)| > 0$, and $|k_2| > |k_1| > 0$, and $|\hat{f}(k)| = 0$ for $k \neq k_1, k_2$. Consider the loss function of $L = L(k_1) + L(k_2)$ with gradient descent training. Denote*

$$\mathcal{S} = \left\{ \frac{\partial L(k_1)}{\partial t} \leq 0, \frac{\partial L(k_1)}{\partial t} \leq \frac{\partial L(k_2)}{\partial t} \right\},$$

*that is, $L(k_1)$ decreases faster than $L(k_2)$. There exist positive constants $c$ and $C$ such that for sufficiently small $\delta$, we have*

$$\frac{\mu\left(\{W : \mathcal{S} \quad holds\} \cap B_\delta\right)}{\mu(B_\delta)} \geq 1 - C \exp(-c/\delta),$$

*where $B_\delta \subset \mathbb{R}^m$ is a ball with radius $\delta$ centered at the origin and $\mu(\cdot)$ is the Lebesgue measure.*

*Proof.* By gradient descent algorithm, we obtain

$$\frac{\partial L(k_1)}{\partial t} = \sum_{l,j} \frac{\partial L(k_1)}{\partial \theta_{lj}} \frac{\partial \theta_{lj}}{\partial t}$$

$$= -\sum_{l,j} \frac{\partial L(k_1)}{\partial \theta_{lj}} \frac{\partial(L(k_1) + L(k_2))}{\partial \theta_{lj}}$$

$$= -\sum_{l,j} \left(\frac{\partial L(k_1)}{\partial \theta_{lj}}\right)^2 - \sum_{l,j} \frac{\partial L(k_1)}{\partial \theta_{lj}} \frac{\partial L(k_2)}{\partial \theta_{lj}},$$

$$\frac{\partial L(k_2)}{\partial t} = -\sum_{l,j} \left(\frac{\partial L(k_2)}{\partial \theta_{lj}}\right)^2 - \sum_{l,j} \frac{\partial L(k_1)}{\partial \theta_{lj}} \frac{\partial L(k_2)}{\partial \theta_{lj}},$$

and

$$\frac{\partial L}{\partial t} = \frac{\partial(L(k_1) + L(k_2))}{\partial t} = -\sum_{l,j} \left(\frac{\partial L(k_1)}{\partial \theta_{lj}} + \frac{\partial L(k_2)}{\partial \theta_{lj}}\right)^2 \leq 0. \tag{43}$$

To obtain

$$0 < \frac{\partial L(k_1)}{\partial t} - \frac{\partial L(k_2)}{\partial t} = -\sum_{l,j} \left[\left(\frac{\partial L(k_1)}{\partial \theta_{lj}}\right)^2 - \left(\frac{\partial L(k_2)}{\partial \theta_{lj}}\right)^2\right], \tag{44}$$

it is sufficient to have

$$\left|\frac{\partial L(k_1)}{\partial \theta_{lj}}\right| > \left|\frac{\partial L(k_2)}{\partial \theta_{lj}}\right|. \tag{45}$$

Eqs. (43, 44) also yield to

$$\frac{\partial L(k_1)}{\partial t} < 0.$$

Therefore, Eq. (45) is a sufficient condition for $\mathcal{S}$. Based on the theorem 1, we have proved the theorem 2. $\qquad\square$

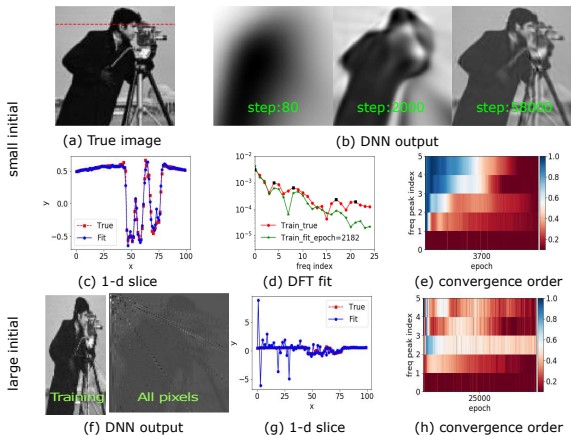

Figure 6: F-Principle in fitting a natural image. The training data are all pixels whose horizontal indices are odd. We initialize DNN parameters by a Gaussian distribution with mean 0 and standard deviation 0.08 (small initial) or 1 (large initial). (a) True image. (b-g) correspond to the case of the small initial parameters. (f-h) correspond to the case of the large initial parameters. (b) DNN outputs of all pixels at different training epochs. (c, g) DNN outputs (blue) and the true gray-scale (red) of test pixels at the red dashed position in (a). (d) $|\hat{h}(k)|$ (green) at certain training epoch and $|\hat{f}(k)|$ (red) at the red dashed position in (a), as a function of frequency index. Selected peaks are marked by black dots. (e, h) $\Delta_F(k)$ computed by the training data at different epochs for the selected frequencies in (d). (f) DNN outputs of training pixels (left) and all pixels (right) after training. We use a tanh-DNN with widths 2-400-200-100-1. We train the DNN with the full batch and learning rate 0.0002. The DNN is trained by Adam optimizer (Kingma & Ba, 2014) with the MSE loss function.

## F    MEMORIZING 2-D IMAGE

We train a DNN to fit a natural image (See Fig. 6(a)), a mapping from coordinate $(x, y)$ to gray scale strength, where the latter is subtracted by its mean and then normalized by the maximal absolute value. First, we initialize DNN parameters by a Gaussian distribution with mean 0 and standard deviation 0.08 (initialization with small parameters). From the snapshots during the training process, we can see that the DNN captures the image from coarse-grained low frequencies to detailed high frequencies (Fig. 6(b)). As an illustration of the F-Principle, we study the Fourier transform of the image with respect to $x$ for a fixed $y$ (red dashed line in Fig. 6(a), denoted as the target function $f(x)$ in the spatial domain). The DNN can well capture this 1-d slice after training as shown in Fig. 6(c). Fig. 6(d) displays the amplitudes $|\hat{f}(k)|$ of the first 40 frequency components. Due to the small initial parameters, as an example in Fig. 6(d), when the DNN is fitting low-frequency components, high frequencies stay relatively small. As the relative error shown in Fig. 6(e), the first five frequency peaks converge from low to high in order.

Next, we initialize DNN parameters by a Gaussian distribution with mean 0 and standard deviation 1 (initialization with large parameters). After training, the DNN can well capture the training data, as shown in the left in Fig. 6(f). However, the DNN output at the test pixels are very noisy, as shown in the right in Fig. 6(f). For the pixels at the red dashed lines in Fig. 6(a), as shown in Fig. 6(g), the DNN output fluctuates a lot. Compared with the case of small initial parameters, as shown in Fig. 6(h), the convergence order of the first five frequency peaks do not have a clear order.

