# OpenReview forum: "Frequency Principle: Fourier Analysis Sheds Light on Deep Neural Networks"
_ICLR.cc/2020/Conference — Reject_

### Official Review · AnonReviewer1 · 2019-10-22
**Official Blind Review #1**

**Rating:** 6

**Review:**

This paper proposes to analyze the loss of neural networks in the Fourier domain. Since this is computationally expensive for larger-dimensional datasets, the analysis instead first projects the data onto the principal component of the data, and then using a Gaussian kernel estimation (which has nice properties in the Fourier domain). The analysis finds that DNNs tend to learn low-frequency components before high-frequency ones.

Overall I quite like the analysis of this paper. I think it could be clearer and contain more experiments but it is otherwise rather convincing proof that DNNs learn low-frequency patterns first.


- More experiments: in particular, analyzing this phenomenon over more than a single principal component or through non-linear transformations of the data.
- It's not always clear how $\mathbf{k}$ is calculated or where it comes from, whether it is implicit through the Gaussian metric or chosen randomly. The paper would benefit from always making this clear in the text and figure captions.
- It's well known that different optimizers seem to learn differently both in terms of speed and features that end up being learned (thus generalization), repeating this analysis for Adam, RMSprop & friends would be great.

The paper is mostly easy to read, but there are a few mistakes here and there that slow down reading. Here are a few:
- "variation problems" you mean "variational"?
- "This difference implicates" -> "implies"
- "by the Parseval's theorem" -> "by Parseval's theorem" (occurs multiple times)
- "difference of" -> "difference between"
- "in previous section" -> "in the previous section"
- "to verify F-Principle" -> "to verify the F-Principle"

**Experience Assessment:**

I have read many papers in this area.

**Review Assessment: Checking Correctness Of Derivations And Theory:**

I did not assess the derivations or theory.

**Review Assessment: Checking Correctness Of Experiments:**

I assessed the sensibility of the experiments.

**Review Assessment: Thoroughness In Paper Reading:**

I read the paper at least twice and used my best judgement in assessing the paper.

---

> ### Author Response · Authors · 2019-11-07
> **To Reviewer 1**
>
>
> We thank you for very helpful and detailed comments. We respond to your concerns as follows.
>
> 1 We had also looked at other directions, e.g., second or higher principle component, and found that F-Principle always holds. We will add more details about these results in our revision.
>
> 2 In all our experiments, we consistently consider the response frequency of $\{\vec{y}_i\}_{i=0}^{n-1}$ on training inputs $\{\vec{x}_i\}_{i=0}^{n-1}$  by the standard nonuniform discrete Fourier transform (NUDFT). See more in Authors' comment [RFD]. For your convenience, we briefly respond to this concern as follows. In the projection method, each frequency k is computed by the NUDFT. Due to the space limit, we make these formulas inline. In the filtering method, we do not compute each frequency due to the high computational cost of high-dimensional Fourier transform. We use a Gaussian filter to decompose the frequency domain into a low-frequency part and a high-frequency part, and then examine the F-Principle in the radially averaged sense. We will make the response frequency clearer in the revision.
>
> 3 We did experiments for various optimizers including these suggested by the reviewer and even non-gradient based methods. F-Principle can always be robustly observed. We will add discussions about these results in our revision.
>
> Finally, thank the reviewer for improving our writing. Please do not hesitate to tell us if you have more concerns.

---

### Official Review · AnonReviewer3 · 2019-10-23
**Official Blind Review #3**

**Rating:** 3

**Review:**

The paper studies the training process of NNs through the lens of Fourier analysis. The authors argue that during the training process, NNs will first learn low frequencies part of the function first and then the high frequency part. To verify this claim empirically, the author propose two methods: 1. examine the convergence of different frequencies in a pre-selected direction in the frequency space during training; 2. examine the convergence rate of the 2-norm of low v.s. high frequencies during training.  Through the experimental results of these two methods, the authors conclude that NNs learn the low  frequency components before the high frequency components. The authors also discuss a potential application of this observation to solving high dimensional PDEs: coupling DNNs training (good at learning low frequency components) with the Jacobi method (good at learning high frequency components). Finally, the authors also provide some theoretical intuition (Thm 1., 2.) why low frequency components are learned faster and an explanation why NNs could generalize well on images but perform poorly on tasks like learning parity functions.


Other comments:
1. It seems the filtering method is a better (might be a sufficient) way to justify the F-Principle than the projection method, given the projection method examines only one direction (also appointed out in the paper).
2. When talking about Fourier transform, would you specify what is the domain of the functions and how the functions are defined (section 3.1) The notation there is somewhat confusing (which makes the rest of the paper difficult to follow) since you are mentioning the Fourier transform of the set {(x_i, y_i)}. It will be helpful to define the function before defining its Fourier transform.  Please also mention what is the domain of the function, {x_i}_i or R^d?
3. According to equation (4), it seems the domain of the functions is {x_i}_i, otherwise equation (4) should be a function of x\in R^d, not x_i.
4. Could you elaborate why (4) is a good approximation of the low frequency energy rather than the L2 norm (over x\in R^d) of (4) with x_i replaced by x\in R^d.
5. It might be useful to refine the related work section. It is not clear what are the previous contributions prior to this paper, and it seems [1] shares some similar results/observation with the this paper.

Overall, I lean to a weak rejection. The key findings (and similar results, e.g. NNs learn simple functions first), i.e. F-Principle seems to have already appeared in previous works, e.g. [1] and the theoretical results of this paper are limited to an idealized setting (results of more general setting appear in another work, mentioned in the paper.)


[1]Nasim Rahaman, Devansh Arpit, Aristide Baratin, Felix Draxler, Min Lin, Fred A Hamprecht,
Yoshua Bengio, and Aaron Courville. On the spectral bias of deep neural networks. arXiv preprint
arXiv:1806.08734, 2018. 1, 8, A




**Experience Assessment:**

I do not know much about this area.

**Review Assessment: Checking Correctness Of Derivations And Theory:**

I assessed the sensibility of the derivations and theory.

**Review Assessment: Checking Correctness Of Experiments:**

I assessed the sensibility of the experiments.

**Review Assessment: Thoroughness In Paper Reading:**

I read the paper at least twice and used my best judgement in assessing the paper.

---

> ### Author Response · Authors · 2019-11-07
> **To reviewer 3**
>
> Thanks for helpful comments. We respond to your concerns as follows.
>
> (1) The filtering method is our important contribution  since high dimension is a key feature of real data.
>
> Your concerns in (2,3,4) can be clarified by the definition of response frequency based on the standard non-uniform discrete Fourier transform (NUDFT)  in Authors' comment [RFD]. For your convenience, we respond to each concern in detail.
>
> (2) A short answer is we CANNOT define the function. A long one is as follows.  (See wiki for more) When dealing with real data, we do not know the true function, which is defined on $\mathbb{R}^d$. Therefore, we CANNOT define the function since we only have some FIXED training samples $\{\vec{x}_{i},\vec{y}_{i}\}$, where $\vec{x}_i \in R^d$. "As a generalized approach for nonuniform sampling, the NUDFT allows one to obtain frequency domain information of a finite length signal at any frequency."
>
> (3) The Eq. (4) can be performed on all points in $R^d$, however, since we have only limited training data points in real applications, Eq. (4) is performed discretely on $\{\vec{x}_{i},\vec{y}_{i}\}$.
>
> (4) The $L^2$ norm (over $\vec{x} \in R^d$) is the total energy, including all frequencies (Parseval theorem). Next, we explain why Eq. (4) is a good approximation of the low-frequency part. Eq. (4) is a convolution. Fourier transform (FT) of convolution in the spatial domain is equivalent to the multiplication of Fourier transforms in the frequency domain (see convolution theorem in \url{https://en.wikipedia.org/wiki/Convolution_theorem}). Therefore, the FT of Eq. (4) equals to the FT of $y$ multiplied by the FT of Gaussian function. Since, the FT of a Gaussian is still a Gaussian, which exponentially decays w.r.t. frequency, the FT of Eq. (4) almost loses all high-frequency information of y. This is why Gaussian is a low-frequency filter. As a simple example, when the variance of the Gaussian filter goes to infinity, Eq. (4) leads to that $\vec{y}^{{\rm low},\delta}_{i}$ equals to the mean (zero frequency) of y for all $i$. When the variance goes to zero, the Gaussian tends to be a delta function, $ \vec{y}^{{\rm low},\delta}_{i}=\vec{y}_{i}$ for each $i$, i.e., keeping information of all frequencies. Therefore, by tuning the Gaussian variance, we can control how much low-frequency information is kept in ${ \vec{y}^{{\rm low},\delta}_i}$.
>
> (5) We would refine our related work section to state our contribution clearer. In the following, we detail our contribution compared with previous works.
>
> Empirical results:
>
> F-Principle was first discovered in [1] and [2] simultaneously through simple synthetic data and not very deep networks. (We use the name F-Principle following work [2]) Although in the revised version of [1], they also examine the F-Principle in the MNIST dataset. However, they add artificial noise to MNIST, which CONTAMINATES the labels and damages the structure of real data.
>
> However, in deep learning, empirical phenomena VARY from one network structure to another, from one dataset to another. Most importantly, they may exhibit significant difference between synthetic data and high dimensional real data. In addition, networks in real applications are very deep while those in [1,2] are not very deep. Therefore, it is important to empirically verify F-Principle in more general and realistic settings.
>
> Our contribution in the empirical verification is as follows: (a) Our paper utilizes two methods to verify the F-Principle in high-dimensional dataset (MNIST and CIFAR10) from different perspectives. (b) We show F-Principle holds in very deep networks (VGG16). (c) We show F-Principle holds in both CNN and fully connected networks. (d) We show F-Principle holds not only for previously investigated MSE Loss, but also for cross-entropy loss in classification problems and variational loss function in solving PDE. Overall, this work provides a convincing empirical demonstration for the universality of the F-Principle that DNNs learn low-frequency patterns first.
>
> Theoretical results:
>
> In this paper, we prove the F-Principle in an ideal setting. This is important because (a) it provides key insights into the mechanism underlying the F-Principle; (b) it inspires a more rigorous proof in a following work, which is mentioned in the paper as mentioned by the reviewer. According to the public dates of the papers in arxiv, this paper is much earlier than that rigorous theoretical paper. The rigorous theoretical paper also clearly states that its idea FOLLOWS this under-reviewed paper.
> The theoretical study of the gradient of $\tanh(x)$ in the Fourier domain is adopted by [1], in which they generalize the analysis to ReLU and show similar results.
>
> Please do not hesitate to tell us if you have more concerns.
>
> [1] Nasim Rahaman et al. On the spectral bias of deep neural networks.  arXiv:1806.08734v2
>
> [2] Xu et al. Training behavior of deep neural network in frequency domain.  arXiv:1807.01251

---

### Official Review · AnonReviewer2 · 2019-10-23
**Official Blind Review #2**

**Rating:** 3

**Review:**

This paper mainly focuses on experimental results on real data to verify the so-called Frequency Principle: DNNs often fit target functions from low to high frequencies during the training process. Some theoretical analyses are also provided to backup the empirical observation.

This paper is very well-written. The methods are explained very clearly, and the logic is easy to follow. However I think this paper also has some weak points, as listed below:

(1) The frequency principle lacks a rigorous definition. In Section 2, the authors provide very inspiring explanations and examples, however no rigorous definitions are given. Is there a way to directly quantitatively define the response frequency?

(2) In Section 3.1, it is not explained why the frequencies are calculated based on the samples. Probably I have missed something, but based on the description, can’t the frequencies be directly calculated on a grid along the 1-dimensional subspace defined by the mean and first principle component of the data? In other words, even after training the predictor function with real data samples for several steps, the frequency of a predictor function should still be its own property and  should be independent of the distribution of data inputs. In fact, are the vectors $n^{-1/2} (cos( 2\pi k x_{p_1,1}), cos( 2\pi k x_{p_1,2}), \ldots, cos( 2\pi k x_{p_1,n}) )$ for different $k$ even orthonormal vectors? I think unless $x_{p_1,i}$ follows certain specific distributions, these vectors are not even close to orthonormal. Therefore using them to calculate the frequencies is very weird.

(3) In fact, are Sections 3,4 and 6 studying the same kind of frequency? It is not very clear due to the vague definitions.

Because of these concerns, I think this paper is on the borderline. For now I tend to recommend a weak reject.


**Experience Assessment:**

I have published one or two papers in this area.

**Review Assessment: Checking Correctness Of Derivations And Theory:**

I assessed the sensibility of the derivations and theory.

**Review Assessment: Checking Correctness Of Experiments:**

I assessed the sensibility of the experiments.

**Review Assessment: Thoroughness In Paper Reading:**

I read the paper at least twice and used my best judgement in assessing the paper.

---

> ### Author Response · Authors · 2019-11-07
> **To Reviewer #2**
>
>
> Thank you for insightful comments. We response to your concerns as follows.
>
> (1) The rigorous definition of the F-Principle relies on the rigorous definitions of response frequency and converging speed. For your convenience,  we recapitulate detailed definition and calculation of response frequency in Authors' comment [RFD] based on non-uniform discrete Fourier transform (NUDFT), which is a standard technique for the evaluation of frequency for nonuniformly sampled data. And the converging speed is defined by the relative error (Sec. 3.1 and 4.1). Therefore, the definition of the F-Principle is defined rigorously. We will elaborate this definition clearer in the revisions.
>
> (2) Your concern can be solved by clarifying the definition of response frequency, which can be found in Authors' comment [RFD]. Here we also explains it in more details. First, in the real datasets, the frequencies CANNOT be directly calculated on a grid along the $1$-dimensional subspace. This is because we DO NOT know the ground truth. We only have some FIXED training data $\{\vec{x}_i,\vec{y}_i\}$, which are non-uniformly sampled. The response frequencies are calculated by the standard NUDFT on dataset $\{\vec{x}_i, \vec{y}_i\}$, which depend on both the input $\{\vec{x}_i\}$ and output $\{\vec{y}_i\}$. Second, although during the training process $\{\vec{x}_i\}$ does not change, the output of the predictor function (here is DNN) keeps evolving. Therefore, it is not weird to compute the response frequencies. Third, to avoid confusion, in Section 2, we emphasize that the response frequency of $\{\vec{y}_i\}$ w.r.t. the input $\{\vec{x}_i\}$ is considered.
>
> (3)  All Sections 3, 4 and 6 contributes to the study of the response frequency. The only difference is that we apply NUDFT for experimental estimation of frequency on real datasets whereas we apply the original integral form of Fourier transform for our theorems. We would emphasize the response frequency more in the revision.
>
> Finally, we hope the clarification of NUDFT in Authors' comment [RFD] can help you make more sense of this paper. Please do not hesitate to tell us if you have more concerns.

---

### Public Comment · ~Rui_Wang1 · 2019-11-03
**Results on ImageNet?**

As the title suggests, have you guys tried on your analysis on ImageNet?

---

> ### Author Response · Authors · 2019-11-07
> **Done on small portion of ImageNet**
>
> Thanks for your suggestion. We did some test on small portion of ImageNet. The F-Principle holds. However, we yet to perform a large-scale experiment.

---

### Author Response · Authors · 2019-11-07
**[RFD] Response frequency definition**

Response frequency of training data $\{\vec{y}_i\}_{i=0}^{n-1}$ on inputs $\{\vec{x}_i\}_{i=0}^{n-1}$.


In all our experiments, we consistently consider the response frequency defined for the mapping function $g$ between inputs and outputs, say $\mathbb{R}^d\to\mathbb{R}$ and any $\vec{k}\in\mathbb{R}^d$ via the standard nonuniform discrete Fourier transform (NUDFT)
$$
\hat{g}_{\vec{k}}= \frac{1}{n}\sum_{i=0}^{n-1}{g(\vec{x}_{i})\mathrm{e}^{-\mathrm{i} 2\pi\vec{k}\cdot\vec{x}_i}},
$$
which is a natural estimator of frequency composition of $g$. (More details can be found in \url{https://en.wikipedia.org/wiki/Non-uniform_discrete_Fourier_transform}.) As $n\to \infty$, $\hat{g}_{\vec{k}}\to\int{g(\vec{x})\mathrm{e}^{-\mathrm{i} 2\pi\vec{k}\cdot\vec{x}}\nu(\vec{x}){\ \mathrm{d}\vec{x}}}$, where $\nu(\vec{x})$ is the data distribution.
We restrict all the evaluation of Fourier transform in our experiments to NUDFT of $\{\vec{y}_i\}_{i=0}^{n-1}$  at $\{\vec{x}_i\}_{i=0}^{n-1}$ for the following practical reasons.

(i) The information of target function is only available at $\{\vec{x}_i\}_{i=0}^{n-1}$ for training.

(ii) It allows us to perform the convergence analysis. As $t\to \infty$, in general, $h(\vec{x}_i,t)\to \vec{y}_{i}$ for any $i$ ($h(\vec{x}_i,t)$ is the DNN output), leading to $\hat{h}_{\vec{k}}\to \hat{y}_{\vec{k}}$ for any $\vec{k}$. Therefore, we can analyze the convergence at different $\vec{k}$ by evaluating $\Delta_{F}(\vec{k})=|\hat{h}_{\vec{k}}-\hat{y}_{\vec{k}}|/|\hat{y}_{\vec{k}}|$ during the training. If we use a different set of data points for frequency evaluation of DNN output, then $\Delta_{F}(\vec{k})$ may not converge to $0$ at the end of training.

(iii) $\hat{y}_{\vec{k}}$ faithfully reflect the frequency structure of training data $\{\vec{x}_i,\vec{y}_i\}_{i=0}^{n-1}$. Intuitively, high frequencies of $\hat{y}_{\vec{k}}$ correspond to sharp changes of output for some nearby points in the training data. Then, by applying a Gaussian filter and evaluating still at $\{\vec{x}_i\}_{i=0}^{n-1}$, we obtain the low frequency part of training data with these sharp changes (high frequencies) well suppressed.

In practice, it is impossible to evaluate and compare the convergence of all $\vec{k}\in\mathbb{R}^d$ even with a proper cutoff frequency for a very large $d$ of $O(10^2)$ (MNIST) or $O(10^3)$ (CIFAR10) due to curse of dimensionality. Therefore, we propose the projection approach, i.e., fixing $\vec{k}$ at a specific direction and the filtering approach as detailed in Section 3 and 4, respectively.

---

### Decision · Program_Chairs · 2019-12-19

**Decision:**

Reject

**Comment:**

Borderline decision.  The idea is nice, but the theory is not completely convincing.  That makes the results in this paper not be significant enough.